# Examining the impact of social media usage on start-ups performance: Mediating role of brand image

**Emmanuel Bruce**[1]*, **Zhao Shurong**[2]*, **John Amoah**[3], **Sulemana Bankuoru Egala**[4], **Philip Adu Sarfo**[5], **Bernard Ekow Baidoo**[1], **Dennis Akomanyi Darko**[1], **Luo Ailing**[2], **Ynag Yongxing**[2]

**1** School of Management and Economics, University of Electronic Science and Technology of China, Chengdu, China; Center for West Africa Studies of UESTC, Chengdu, China, **2** School of Public Administration, University of Electronic Science and Technology of China, Chengdu, China; Center for West Africa Studies of UESTC, Chengdu, China, **3** Department of Marketing and Human Resource Management, KAAF University College, Kasoa, Ghana, **4** Department of Informatics, Faculty of ICT, SD Dombo University of Business and Integrated Development Studies, Bamahu, Ghana, **5** School of Management Engineering, Zhengzhou University, Zhengzhou, China,

* kinbuki100@outlook.com (EB); shurz2015@163.com (ZS)

## Abstract

Social media has emerged as an assertive communication and brand-building tool in the dynamic entrepreneurship landscape. This study explores the influence of social media usage on the performance of start-ups, focusing on the mediating role of brand image. The research employs a quantitative approach, collecting data from 450 start-ups in Ghana through surveys. Data collected was processed and analyzed through PLS-SEM. The findings of the study supported all the formulated hypotheses. The outcome suggests that, social media, brand image and innovation capabilities all have direct and positive linked with startup performance. Additionally, the findings proved a mediating role of brand image between social media usage and startup performance. Understanding the dynamics between social media, brand image, and performance is vital for start-ups seeking to thrive in competitive markets. Based on the outcomes in the findings, the study recommends social media marketing tool for startup businesses in Ghana to stern competition and drive sustenance. This research contributes to academic literature and practical insights, offering nuanced perspectives on leveraging social media as a strategic tool for cultivating a brand image and influencing overall start-up performance. The implications of the study findings serve as guideline for startups and young entrepreneurs' in developing countries as they develop their marketing strategies.

## Introduction

Small businesses, irrespective of size, have distinctive characteristics. These features have been highlighted in the works of [1], which state that startup entrepreneurial activities expand innovative strategies. Startups' significance to economic growth includes creating employment and launching new products [2,3], generating higher returns, and alleviating socio-economic

**Data availability statement:** https://figshare.com/articles/journal_contribution/Social_Media_Usage_and_Performance_of_Start-ups/26296315

**Funding:** This work was supported by the following funding sources: Sichuan Social Science Planned Key Research Project (No. SCJJ23ND61) (originally included); 2022 Central Chinese University Fundamental Research Program for Humanities and Social Science Cultivation Key Project (No. ZYGX2022FRJH004). Regional Studies of the Ministry of Education of China (No. 2024-N01).

**Competing interests:** The authors have declared that no competing interests exist.

inequalities [4,5]. Startup entrepreneurial activities have several social, environmental, and economic impacts [6]. Such immense contribution confirmed the need for government investment in digitalization in developing economies since startups and small businesses cannot cope with the recent globalization process [7,8]. Consequently, passage to digitalization by startups is crucial to their sustenance since digitalization has become an essential resource for contemporary businesses [9].

There is a current and growing interest on the part of contemporary businesses including startup to invest in the emerging sophisticated tool such as social media in general, and particularly in areas marketing [10,11] and customer satisfaction [12–14]. As summarized by [15–17] for instance, young entrepreneurs have started utilizing social media to facilitate effective communication with consumers and business visibility. As a result, social media adoption can lead to double profits, which boosts marketing effectiveness [4]. Moathen and Almaktoon [2] and Troise et al. [16] strengthens the research that, the adoption and utilization of social media has improved the marketing performance of start-ups. Besides, Troise et al. [2] echoed that, value co-creation, entrepreneurial orientation and opportunities can be created through the utilization of social media. Although social media leverage is emphasized, start-ups and small enterprises frequently struggle with a lack of awareness and comprehension of how to use social media tools [18–20]. Therefore, it is critical to comprehend how social media affects startups' performance. As Abane et al. [21] demonstrated, recent studies offered significant empirical support for the hypothesis concerning the influence of social media usage on startup performance. Their research shows that social media strategy significantly impacts startup performance, with the brand image as a key mediator in this relationship.

Furthermore, [22,23] illustrates how, through digital platforms, social media tools have been able to empower local entrepreneurs, which is quite evident among startups in Ghana benefiting from improved business performance and enhanced brand image. Abane [21] also opined that start-up utilization enhances entrepreneurial orientation, improving business performance and enhancing customer satisfaction. Thus, social media has become an effective and economical medium to access a vast potential customer pool [2,14], strengthen brand recognition [6], interact with customers directly [5,7], collect important market information [13] and ultimately increase sales and growth [12,24]. However, other scholars have suggest that social media strategy is crucial for addressing startup challenges such as limited resources (i.e., financial challenges and limited advertising strategy [6,10]. Additionally, tackling the startup challenges through social media adoption will likely positively affect business performance.

As the startup ecosystem evolves, young entrepreneurs must adopt and utilize modern technologies such as social media technologies to compete and achieve superior advantage [3,5]. Despite significant studies highlighting effective strategies and essential aspects of social media marketing for managing businesses in today's era of digitalization, there remains limited research on how small enterprises and start-ups can optimize their use of social media and enhance brand image effectively. Moreover, only a handful of startups have attained successful outcomes since numerous encountered difficulties in developing strategies to leverage social media's potential in the context of Ghana [4,22,25]. Moreover, although scholars have confirmed the significant effect of social media on startups and small businesses [e.g., 26,27], others have called for examining the mediating role of brand image in the relationship between social media and startup performance [28]. Therefore, this study proposes that the relationship between social media usage and startup performance primarily influences brand image (mediated). Specifically, this study seeks to answer the following research questions: a) What are the effects of social media usage on brand image and startup performance? b) how do innovation capabilities affect startups' performance? 3) how does a brand image

mediate the relationship between social media usage and startup performance? Addressing these questions will not only enhance our understanding of this phenomena but also inform policy and strategic decisions that can optimize the social media strategy as a marketing tool, offering avenue for enhancing brand image in the context of startups in Ghana.

Hence, the aim of this study was to investigate the impact of social media usage on start-ups performance, identifying the mediating role of brand image. The present study intended to provide actionable insights for young entrepreneurs, policymakers, businesses, and investors seeking to leverage on social media marketing to enhance business performance. The paper structure is arranged as follows: Section 2 presents the theoretical background, literature review, and hypothesis development of the study; Section 3 focuses on the research methodology of the study; Section 4 presents the outcome of empirical findings and analysis; and lastly, Section 5 presents both theoretical and practical implications and conclusion of the study.

## Review of literature

### Theoretical background

This paper applied the resource-based view (RBV) theory proposed by [29]. Per the theory, an *"organization's performance hinged on the possession of superior resources"*. RBV explains the correlation between an organization's internal characteristics and performance [29]. Shin et al. [30] argued that possessing resources alone cannot help firms achieve a competitive advantage. Still, the firms also can acquire, combine, and deploy valuable resources to achieve strategic actions [31,32]. Lonial and Carter [33] also highlighted the effective implementation of resources to attain strategic actions, which leads to sustained superior advantage. In this context, social media is the firm's ability to utilize this technology and other valuable assets to achieve organizational goals and improve performance. Regarding improving business performance through social media usage, [34] submits that technological innovation (i.e., social media) can be utilized together with firms' other resources to enhance performance since integration and deployment of resources aid in organizational practices.

Social media uniquely provides companies with fresh opportunities for innovation, which can result in enhanced credibility, success, and sustainability [35–38]. In investigating social media usage and young entrepreneur investment, Chen and Liu [35] revealed how social media has become an important marketing tool for entrepreneurship activities. In this regard, start-ups can adopt and utilize social media marketing as a resource strategy to create brand awareness and to communicate with the audience at each start-up lifecycle [9,39]. Drawing on resource-based view theory, Onngam and Charoensukmongkol [36] investigated the impact of social media capabilities on the performance of small and medium enterprises, focusing on firms in Thailand. Moreover, other researchers [31,35,40] have employed the research-based view theory to investigate the impact of social media on start-up performance and sustainability. In this paper, the application of resource-based view theory would an insight on the adoption and utilization of social media by startups' as a resource to drive sustainability in developing countries, particularly Ghana, as few empirical works have applied this theory. The present study envisage that start-ups with higher usage of social media marketing capabilities have the potential to effectively achieve superior performance [16,41].

### Start-ups in Ghana

According to Schumpeter [42], entrepreneurial activities contribute to a nation's economic performance by introducing innovative products and enhancing productivity and competition in the broader context. As Johnson [25] mentioned, technological innovation supports the

entrepreneurship ecosystem, which drives economic growth, especially in developing nations. For instance, technology empowers entrepreneurs to create value, innovate and acquire customers for startups. It is also important to note that technology has played a crucial role in startup business transition, offering the essential tools and platforms to establish and expand [43–46]. Mercandetti et al. [47] highlighted the critical role of start-up ecosystems (i.e., entrepreneurs, accelerators, investors, financial service providers, and government agencies) in entrepreneurial activities. These ecosystems foster local economic vibrancy and national economic expansion by creating conditions for emerging and expanding businesses to succeed [48–50]. Regardless, entrepreneurs in emerging economies require an enabling environment and policy and business environment to survive.

In Ghana and other developing nations, *"entrepreneurship – and entrepreneurship policy is not merely about small business, or even at times about business at all – is about creating environments in which people can perceive entrepreneurial opportunities, opportunities to improve their lives and to be empowered by the environment to act upon their visions"* [51]. Based on this, governments from developing economies are embarking on creating entrepreneurial start-ups, supported by policies and private sector initiatives [51] and encouraging young graduates to venture into entrepreneurial activities [45,52]. For instance, the Ghanaian Government established the National Entrepreneurship and Innovation Plan (NEIP) to boost the local entrepreneurship ecosystem, mainly to entice young graduates to engage in start-up entrepreneurial activities. Furthermore, institutionalized resources, including incubators, innovation hubs, and networking events accelerators, have been provided by the government of Ghana to expand the start-up businesses [53,54], investing over $100m in the NEIP and pledged to offer more support when necessary [55].

Start-up businesses have expanded in developing economies like Ghana due to the recent advancement of information technology [56]. Okrah et al. [57] focus on factors of startup success and growth by arguing that social media technologies are taking over the Ghanaian business landscape to drive sustenance, especially among small and medium enterprises. Several works have established that social media enables startups in Ghana to connect the entire ecosystem [58–60], sharing knowledge [61] and networking [62].

## Social media usage and start-ups performance

Social media has become a strategic marketing platform to engage and grow audiences, increase brand awareness, influence purchasing behavior, and generate value for consumers [63,64]. [65] also observed: *"social media marketing as a medium to convey electronic word-of-mouth about a product, brand or company"*. Meanwhile, businesses have been recognized for using social media to generate new business prospects and strategic management approaches, enhancing organizational efficiency by reorganizing their current resources and practices [31]. However, per [66] studies, only 44% of small businesses utilize social media marketing as a communication tool. In this view, it is proposed that startups can effectively adopt social media tools to reach the masses, build brand reputation, communicate pricing strategy, and drive sales [67,68], leading to improved performance.

Past studies (e.g., [53,69,70]) also have established that social media usage is a practical approach to innovation and business performance of young entrepreneurs and startups. The current study focused on social media usage, as an essential tool for entrepreneurial business activities [4,21]. Additionally, [31] hypothesized that social media usage has a positive influence on a small businesses' performance and sustainability. Therefore, we proposed that social media usage positively influences startup performance.

H1. Social media usage is positively associated with Startup Performance

## Social media usage and brand image

According to [71], brands have direct communication with individual consumers. Social media marketing engenders online interaction, creating and reinforcing brand loyalty and trust. It has also been stated that a social marketing strategy is an effective channel for growing a brand and a startup business [72]. Prior theories and empirical studies have established that social media serves as a brand communication tool [73–76]. Owualah [52] for instance, illustrates how, social media significantly affect brand image.

Hilong [17] further asserts that interaction via social media networks is essential in building brand image and loyalty. Their study investigated the effectiveness of social media on the performance on start-up business. The authors proved that social media influence behavioral brand image intention measurements. Additionally, [77] supported the hypothesis that social media positively linked with startups' brand image. Accordingly, the following hypothesis is proposed:

H2: Social media usage positively affects the brand image of start-ups.

## Innovation capabilities and start-up performance

Start-ups are regarded as *"high-tech enterprises with higher R&D intensity and potential to achieve a higher financial performance"* [78]. Innovation capabilities can be categorized into marketing, technology and knowledge, and marketing capabilities [79,80]. Recently, firms have relied on innovation capabilities (i.e., technology, knowledge, and marketing) for growth and sustenance. In this view, [31] highlighted the significance of innovation capabilities in handling market situations among SMEs. For instance, innovation capabilities in organizations drive start-up business growth, increased competitiveness, and increased sales and profitability.

In the context of start-ups, innovation capabilities, especially technology, can be evaluated based on the other valuable assets available and the enterprise's ability to combine the resources and technology in the organizational processes, leading to improved performance [78]. Their study examined innovation capabilities and start-up performance, focusing on South Korea. Prior studies have evidenced the positive impact of technology capabilities on small business growth [81–83]. de Zubielqui and Jones [41] statistically confirmed the positive relationship between innovation and performance, using data from Australia. Based on the above discussion, we, therefore, provide the following hypotheses:

H3: Innovation capabilities will positively affect start-ups' performance

## Brand image and start-up performance

According to [84], brand image is one of the marketing components for enhancing business performance. It has been stated that firms require practical marketing elements (i.e., brand) that can identify or be distinguished from others [49,85]. Hoeffler and Keller [86] also view brand image as a marketing strategy to attract customers. In this vein, enhancing brand image have significant impact on firm performance [87, 88]. Thus, having a strong brand image positively linked to loyal customers, which lead to increased sales and business growth.

In investigating internal branding as a marketing strategy, Ismail et al. [89], aver that social media platforms allow for collaboration with consumers and improve consumer loyalty through a strong brand image. Using comparing lean and the theory-based view, Felin et al. [90] emphasized that startups are "evolving entity" and undergoes a different lifecycle and, thus needs brand identity involvement in each phase of the cycle [91]. Notably,

[92,93] demonstrated that brand image positively affects customer satisfaction and start-up performance. Hence, the study proposed that brand image positively influences startup performance [94–100].

H4: Brand image positively affects start-ups' performance

### Innovation capabilities and brand image

Prior studies have established the importance of innovation capabilities in organizational learning and performance [31,84]. Innovativeness can assist businesses in appreciating the significance of branding, not just for achieving successful commercialization of innovations but also as a valuable resource to introduce new products and services more aligned with customer needs [41]. It is evident that innovations help strengthen the firm's reputation through brand image [80]. Yoo and Kim [82] empirically highlighted that innovativeness significantly affects brand image. The authors confirmed a positive relationship between innovation capabilities and brand image. Yang [81] corroborated their findings, confirming that innovation significantly impacted brand image and corporate growth, particularly brand loyalty and brand image. Furthermore, Borah et al. [31] suggested that innovation positively affects brand image.

H5: Innovation capabilities will positively affect brand image.

## Methods

### Sample procedure

Scholars have recognized the relevance of social media usage in contemporary businesses, especially among small and medium enterprises [83,101–106]. Yet, few studies have been conducted empirically focusing on start-ups in developing economies [41,107]. Henceforth, the present study examines the influence of social media usage on start-ups' performance and the mediating role of brand image in developing economies to fill the knowledge gap. Ghana's start-ups are increasingly expanding due to the support from the government and the interest of young entrepreneurs, mainly graduates [4,45]. Start-ups in Ghana are primarily classified as small and medium enterprises, comprised of more than 90% of businesses in the country [108], contributing about 70% to the Gross Domestic Product [4]. The present study focused on start-up ventures (young entrepreneurs) in Ghana for data collection, employing a random sampling technique, precisely a simple random method. This sampling method was chosen based on its sampling error-free, objectivity, and absence of bias, as mentioned by [109].

The presentation of the structured questionnaire was two-fold, i.e., in sections A and B. The first section (A) focuses explicitly on demographic information, while section (B) covers the study constructs, consisting of twenty-six questions administered to young entrepreneurs owing start-ups in Ghana. Out of the 520 questionnaires administered, 450 of the questionnaires received were valid for data processing and analysis. The questionnaires were administered via Goggle Forms (i.e., through emails and social media platforms such as Facebook Messenger and WhatsApp) and offline methods after formal approval from the selected young entrepreneurs. Additionally, young entrepreneurs were chosen based on those who utilizes social media networks for their business activities and as a marketing communication tool.

Specifically, formal authorization and consent were obtained from the selected young entrepreneurs in Ghana before the start of the data collection processes. To get a certain level of actual data that would be beneficial to both theory and practice, the study utilized young entrepreneurs who have registered their businesses in Ghana under the National Board for

Small-Scale Industries (NBSSI), particularly in Accra and Kumasi to answer the questionnaire based on the comprehensive data that they possessed. It is imperative to note that researchers employed both offline and online methods for data collection, i.e., through social media platforms such as WhatsApp and Facebook and face-to-face interviews for the data collection between 13th September 2023 and 19th December 2023.

Of the questionnaires, 70 were defective, and others were incorrectly filled out for the study's analysis. The researchers eliminated these invalid responses and irregularities [110]. A pilot study was conducted to test the study's constructs' reliability and validity, as indicated by [78]. To add more, the respondents used an average of ten minutes each to answer the questionnaire administered.

The study finally employed Partial Least Squares Structural Equation Modeling (PLS-SEM), version 4.0 technique, for the analysis since it has been widely utilized and applied in social media applications and social sciences [31,111,112]. Additionally, PLS-SEM was used due to its ability to analyze the dependent variables comprehensively, manipulate both measurement and structural models, and precisely estimate mediating effects while reckoning for measurement errors, as indicated by [31].

## Ethical Considerations

**Ethical Approval:** The present study obtained ethical approval from the National Board for Small-Scale Industries (NBSSI) Ethical Committee board in Ghana (SME-ERC/375/09/23) since start-ups in Ghana fall under the small and medium-scale enterprises. The researchers moreover, revealed the study's objectives and other relevant information to the authorities and participants.

**Consent to participate:** The respondents' permission was sought before the collecting the data per the requirement of the National Board for Small-Scale Industries Ghana, participants'. In addition, no money was charged from the respondents for partaking in the survey, and respondents has an option to opt out from the survey anytime.

**Consent for publication:** The authors declare that the human research participants granted informed permission to publish their data.

## The measurement of different variables

To measure the study constructs, the researchers used a five-point Likert scale approach, with one denoting disagree entirely, two disagree, three neutral, four agree, and five completely agree. It is noteworthy to emphasize that most researchers and scholars in the recent past [113,114] have utilized the five-point Likert scale because it provides a means of determining the extent of respondents' opinions and comprehension regarding the subject matter. The following study constructs were modified from earlier research: start-up performance [68,83], brand image [14, 115], innovation capabilities [31,116], and social media usage [16,31]. The ordinal scale was used to measure the constructs. The ordinal scale was selected because it evaluates the respondent's level of agreement or disagreement with the constructs being considered.

## Common method bias

To reduce the effect of common method bias (CMB) in the current study, we employed a dual strategy, mainly procedural and statistical measures. As indicated by [113], CMB exists in self-reported survey data. Therefore, designing and developing a research survey to be free from errors and present it is essential. In this study, the researchers assured the respondents of confidentiality during the data collection period following suggestions by [117].

The researchers first applied Harman's single-factor approach to assess common method variance (CMB) in the study, as put forward by [118]. This test evaluated the CMB of all the constructs with outcomes demonstrating that all items (measurements) in our study model loaded onto 25 distinct factors. The first factor explained 49.776% of the total variance, a favorable outcome although slightly below the expected 50% threshold, as shown in Appendix 2. As established by other scholars [4,119], using a thorough collinearity assessment of Smart PLS, the current study indicated that all variance inflation factor (VIF) values were below the recommended threshold of 5. This, therefore, aver the absence of significant common method bias in our current data for the study.

## Results

### Respondent demographic profile

Per a study conducted by [120], demographic attributes are linked with the methodology chosen in a research study. This study collects demographic information from young entrepreneurs (small and medium business owners) in diverse sectors, including construction, fintech, technology, hospitality, restaurants and foods, clothing and fashion, and retail. As demonstrated by studies by [121] and [122], extant literature emphasizes how crucial it is to comprehend survey respondents' demographic profiles to improve problem-solving. Table 1 presents the demographic data that was collected. It shows that out of the 252 respondents, 56% are males and 44% are females. Bachelor's degree is the most common educational background (38.88%), while others are the least common (7.11%). In industry categories, fashion recorded the highest, with 21.77%. Additionally, 31.33% of young entrepreneurs have less than one year of experience. Lastly, the country's national capital has the highest number of young entrepreneurs, at 29.55%.

### Measurement model assessment

The study further provided insight into the item factor loadings, Cronbach's Alpha, Composite reliability (CR), and Average Variance Extracted (AVE) in Table 2. As recommended by [123] and [124], robust internal consistency, composite reliability, and Cronbach's alpha scores that exceed 0.70 affirm the reliability of the correlations and establish convergent validity (CV) in a particular study. Regarding the outer loadings and measure of convergent validity, values above 0.8 were recorded in the survey. Again, the variance inflation factor (VIF) was analyzed to handle the multi-collinearity concerns, showing VIF values below 5.0, which is the required criterion of multi-collinearity in research, as mentioned by [125]. Moreover, the discriminant validity, demonstrating each construct, was evaluated via Fornell and Larcker's Criterion for dependent variables. This basis was conducted to collate the square root of each AVE construct with its bivariate relationships [126,127], as depicted in Table 3. The first item of Start-Ups Performance was dropped since it falls below the estimated threshold.

The researchers were motivated to use the Fornell-Larcker criteria to assess the discriminant validity of the latent variables by the literature work of [124]. According to [123], Table

**Table 1. Measurements.**

| Variables | Item | Source |
|---|---|---|
| Social Media Usage | 7 | [16] and [31] |
| Brand Image | 7 | [14] and [115] |
| Innovation Capabilities | 6 | [31] and [116] |
| Start-up Performance | 5 | [68] and [83]. |

**Table 2. Profile of respondents.**

| Gender | Items | Frequency | Percentage (%) |
|---|---|---|---|
| | Male | 252 | 56.0 |
| | Female | 198 | 44.0 |
| | Total | 450 | 100 |
| Educational level | Bachelor Degree | 175 | 38.88 |
| | Diploma | 103 | 22.88 |
| | High School Certificate | 64 | 14.22 |
| | Master's Degree/ PhD | 76 | 16.88 |
| | Others | 32 | 7.11 |
| | Total | 450 | 100.0 |
| Industries Categories | Fashion | 98 | 21.77 |
| | Construction | 53 | 11.77 |
| | Technology | 93 | 20.66 |
| | Hospitality | 59 | 13.11 |
| | Hotels/Restaurants | 65 | 14.44 |
| | Others | 82 | 18.22 |
| | Total | 450 | 100.0 |
| Years of Service | Less than a year | 141 | 31.33 |
| | 1-3years | 133 | 29.55 |
| | 3-5years | 61 | 13.55 |
| | More than 5years | 115 | 25.55 |
| | Total | 450 | 100.0 |
| Business Location | Greater Accra | 133 | 29.55 |
| | Western | 93 | 20.66 |
| | Central | 108 | 24.0 |
| | Eastern | 67 | 14.88 |
| | Others | 49 | 10.88 |
| | Total | 450 | 100 |

Source: Field data (September – December 2023).

4 below indicates that every value or figure on the diagonal form exceeded the minimum threshold of 0.5, indicating the Average Variance Extracted (AVE) as a consistent result. The AVE had to have higher values (both in column and row position) than the other constructs, as can be seen in the discriminant validity table below, for the study constructs to meet the requirements of the Fornell-Larcker criteria [126].

**Hypothesis testing - PLS-SEM.** To evaluate the higher-order constructs model (model fit test), the current study employed a bootstrapping technique while utilizing PLS-SEM [123]. Thus, path analysis was done to assess the model fit. The essence of this analysis is to illustrate the causal relationship between the study constructs. This further depicts the significance levels, t-values, and coefficient paths ($R^2$), presenting valuable insights into the relevance of constructs' interrelationships as shown in Fig 1. The researchers moreover, applied advanced mediation procedures to assess mediation analysis as recommended by [125]. Using Smart PLS, the mediation analysis was conducted via a two-step procedure, focusing on evaluating the significant impact of independent variables on dependent variables, as established by other studies [124,126]. Again, the indirect impact of the exogenous variable on the endogenous variable was analyzed, specifically to ascertain the mediating variable.

**Table 3. Evaluation of the validity and reliability of the construct.**

| CONSTRUCT | INDICATORS | LOADINGS | VIF | CA | CR | AVE |
|---|---|---|---|---|---|---|
| Social Media Usage | 1. My enterprise utilizes social media to share information. | 0.738 | 1.742 | 0.896 | 0.865 | 0.553 |
| | 2. My venture uses social media marketing for branding | 0.746 | 1.748 | | | |
| | 3. My enterprise uses social media to receive feedback from customers | 0.747 | 1.725 | | | |
| | 4. My venture uses social media to communicate with stakeholders | 0.774 | 1.807 | | | |
| | 5. Social media usage helps in the introduction of new products and services | 0.720 | 1.645 | | | |
| | 6. My enterprise uses social media to deliver customer services | 0.705 | 1.697 | | | |
| | 7. social media helps to conduct competitive analysis and marketing research. | 0.773 | 1.919 | | | |
| Innovation Capabilities | 1. There is always the introduction of new products and services | 0.780 | 1.790 | 0.851 | 0.890 | 0.573 |
| | 2. My venture uses new technology and knowledge from different resources to search for new information | 0.740 | 1.770 | | | |
| | 3. My enterprise always uses new methods in organizational practices | 0.769 | 1.744 | | | |
| | 4. My firm constantly focuses on innovative ideas from customers and stakeholders | 0.740 | 1.756 | | | |
| | 5. Our enterprise adapts to environmental changes through external resources | 0.763 | 1.832 | | | |
| | 6. There is always innovation in our methods of operation. | 0.750 | 1.702 | | | |
| Brand Image | 1. In my enterprise, social media networks help create brand awareness. | 0.775 | 1.894 | 0.876 | 0.904 | 0.573 |
| | 2. My venture uses social media networks to post pictures, videos, stories, and live videos about a product. | 0.755 | 1.925 | | | |
| | 3. social media is used to build a strong brand identity. | 0.793 | 2.060 | | | |
| | 4. Social media networks offer a medium to engage in conversation regarding products or services. | 0.755 | 1.878 | | | |
| | 5. Brand image helps differentiate my products from other competitors. | 0.769 | 1.941 | | | |
| | 6. Social media aids customers in becoming familiar with a particular brand | 0.725 | 1.679 | | | |
| | 7. Brand image aids start-ups to build trust and loyalty | 0.725 | 1.688 | | | |
| Start-up Performance | 2. Innovation, credibility, and improved performance can be achieved through social media. | 0.831 | 2.083 | 0.848 | 0.892 | 0.623 |
| | 3. Increase in customer loyalty and retention through brand aware-ness can be achieved through social media networks. | 0.797 | 1.784 | | | |
| | 4. Social media enhanced information accessibility and promotion activities | 0.780 | 1.836 | | | |
| | 5. Social media networks help increase sales and transactions through brand image and identity. | 0.807 | 1.921 | | | |
| | 6. Customer Relationship Management can be achieved through social media usage. | 0.726 | 1.596 | | | |

Source: Author's processing from SmartPLS 4.0 software.

Besides, the study measured the statistical significance of coefficients (corresponding t-value >1.96; p-value < 0.05) as recommended by [123]. In this study, the outcome of ($\beta$ = 0.287 and t value = 6.126, p = 0.000), demonstrates a statistically positive and significant relationship between Brand Image and Start-Up Performance, hence H4 is accepted. More-over, the outcome also established a direct relationship between Innovation Capability and Start-Up Performance (with $\beta$ = 0.293, t value = 5.934, p value = 0.000). Furthermore, a

**Table 4. Fornell-Larcker criterion discriminant validity.**

| Constructs | BI | INC | SMU | SP |
|---|---|---|---|---|
| Brand Image | **0.757** | | | |
| Innovation Capability | 0.803 | **0.757** | | |
| Social Media Usage | 0.79 | 0.82 | **0.744** | |
| Start-Up Performance | 0.781 | 0.793 | 0.795 | **0.789** |

Source: processing from SmartPLS 4.0 software.

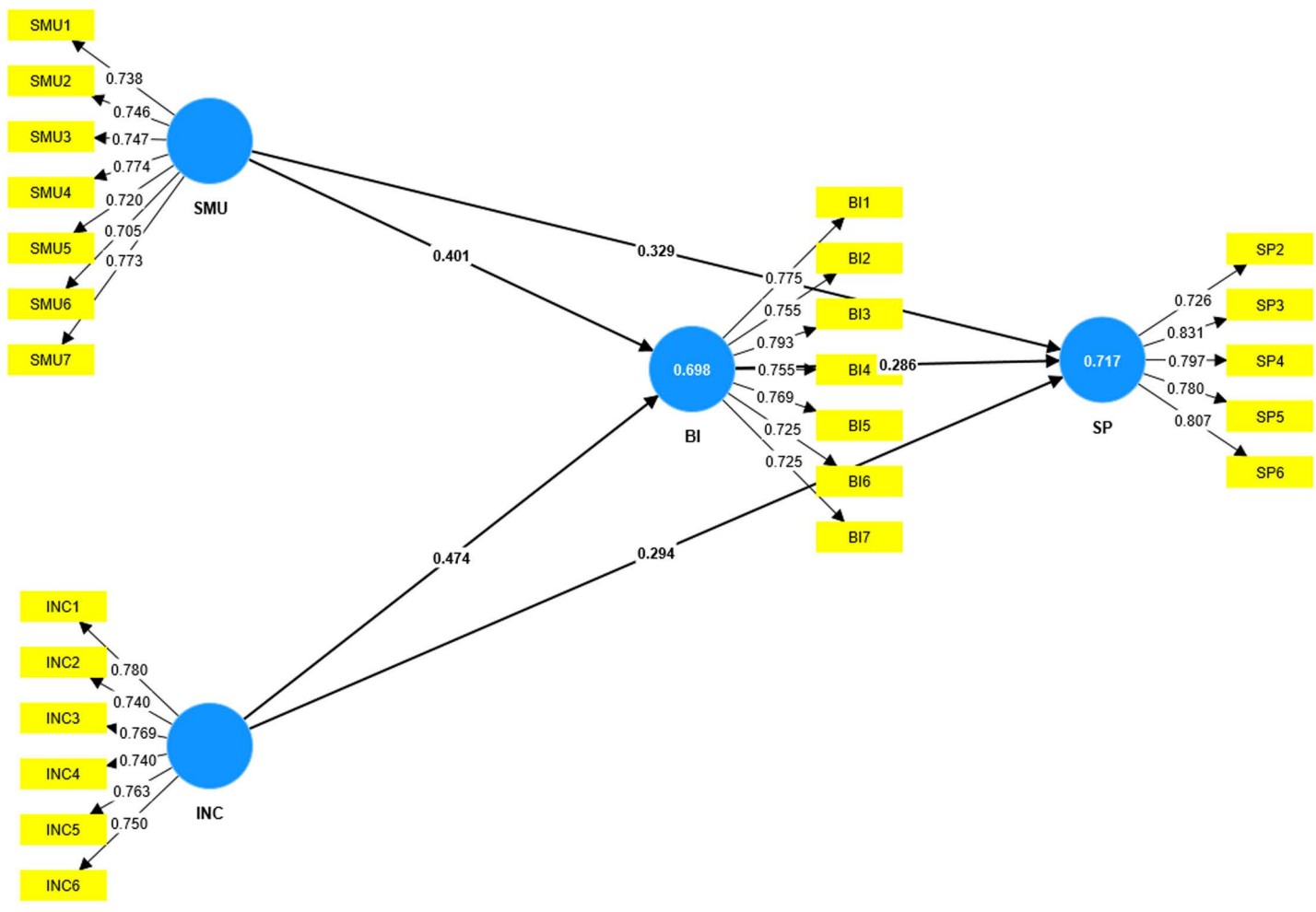

**Fig 1. Measurement model results.**

positive correlation was confirmed among Social Media Usage and Brand Image ($\beta = 0.401$, t value $= 8.787$, p value $= 0.000$) and Social Media Usage and Start-Up Performance with ($\beta = 0.328$, t value $= 6.055$, p value $= 0.000$). Consequently, we concluded that H3, H2, and H1 are acceptable and confirmed.

The study finally evidenced brand image as a mediating variable in the relationship between social media usage and start-up performance ($\beta = 0.115$, t value $= 4.921$, p $= 0.000$); and INC -> BI -> SP ($\beta = 0.136$, t value $= 5.243$, p $= 0.000$) as illustrated in Table 5. The

regression model's coefficient of determination ($R^2$) was assessed about the research model's coefficient of determination, which measures its capacity for prediction. The coefficient indicates the proportion of variance in the dependent variable that can be ascribed to the independent variable, also known as the predictor. Therefore, as Tables 5, 6 and Fig 1 below demonstrate, $R^2$ of 69% for brand image and 68% for start-up performance.

## Discussions of results

This study sought to investigate the effect of social media on a start-up's performance alongside the mediating role of brand image. Given the conventional impact social media have on small businesses, the technology has become a provenance tool for individuals to unearth their talents. Following this, this study proposed that social media will have a positive impact on the entrepreneurial potential of start-ups. Hence, five hypotheses were formulated. As anticipated, all five hypotheses were supported by the study results.

Considering the significant impact of social media as has been confirmed by the literature, the study first hypothesized that, social media usage is positively associated with Start-up Performance was confirmed. Hence, hypothesis H1 was supported consistent with the literature. Supported by [63] and [64], the hypothesis posits that social media serves as a strategic marketing platform. This aligns with the contemporary view of social media as a crucial tool for marketing strategies. A priory, a claim supported by [65] affirmed social media as a medium to convey electronic word-of-mouth, fostering brand awareness. The literature proffers that that social media generates value for consumers, hence, engaging content, customer interaction, and feedback on social media contribute to perceived value. Supported by [31], social media serves as a platform that supports message distribution, sharing various content types, and fostering innovation and development among small businesses. This impression was reinforced by [53,69,70] who agreed that, social media usage is an effective approach for innovation and business performance among young entrepreneurs and start-ups.

Secondly, the study hypothesized that social media usage positively affects the brand image of start-ups which was also supported in line with extant literature. Specifically, [71] conclude that brands can directly communicate with individual consumers through social media. While social media marketing facilitates online interactions that create and reinforce brand loyalty and trust, [72] affirmed that a social marketing strategy is an effective channel for growing

**Table 5. Hypothesis testing.**

| Constructs | Path Co-efficient | Mean | Standard deviation | T statistics | 2.5% | 97.5% | P values | Decision |
|---|---|---|---|---|---|---|---|---|
| H1: SMU -> SP | 0.328 | 0.327 | 0.054 | 6.055 | 0.222 | 0.435 | 0.000 | Supported |
| H2: SMU -> BI | 0.401 | 0.401 | 0.046 | 8.787 | 0.311 | 0.487 | 0.000 | Supported |
| H3: INC -> SP | 0.293 | 0.295 | 0.049 | 5.934 | 0.197 | 0.392 | 0.000 | Supported |
| H4: BI -> SP | 0.287 | 0.286 | 0.047 | 6.126 | 0.196 | 0.379 | 0.000 | Supported |
| H5: INC-> BI | 0.115 | 0.115 | 0.023 | 4.921 | 0.072 | 0.164 | 0.000 | Supported |

Note(s): *p ≤ 0.05; SMU = Social Media Usage; INC = Innovation Capabilities; BI = Brand Image; SP = Start Up Performance. Source: processing from SmartPLS 4.0 software.

**Table 6. Coefficient and predictive relevance.**

| Constructs | $R^2$ | $Q^2$ |
|---|---|---|
| Brand Image | 0.689 | 0.694 |
| Start-up Performance | 0.717 | 0.689 |

both a brand and a startup business. This aligns with the idea that social media plays a vital role in the overall growth and development of start-ups. Other relevant studies (e.g., 74, 75], have affirmed the relationship between social media usage and brand image among start-ups. The collective evidence supports the positive influence of social media on shaping or modifying brand image. [77] provide support for the hypothesis by revealing that social media networks such as Facebook, LinkedIn, and Twitter are effective mediums for customer attraction, brand positioning, and positive linking with start-ups' brand image.

Considering that, innovation capabilities are necessary conditions to sustain innovation, literature has considered it as a critical success factor in entrepreneurship. Hence, the support for hypothesis H3. The support for this hypothesis affirms that innovation capabilities, particularly in technology, positively affect start-up performance and is well-supported by the collective findings of various studies. The evidence suggests that a strong emphasis on technological innovation, along with effective utilization of resources and external support, contributes significantly to the success and performance of start-ups. [78] characterize start-ups as high-tech enterprises with a higher emphasis on research and development (R&D) intensity, indicating a potential for achieving higher financial performance. Additionally, [41] confirmed the positive impact of social media on start-ups' innovation and performance, further reinforcing the broader concept of innovation capabilities, including technology and social media, in driving start-up success. Given Innovation capabilities such as technology, and knowledge, [78] affirmed the role of the government in influencing technological and research and development activities of start-ups. The emphasis on innovation capabilities suggests their importance for growth, competitiveness, increased sales, and profitability [31].

Furthermore, this study sought to investigate how brand image positively affects start-ups' performance. Consistent with the literature, the proposition was supported [74,84]. Affirming, [74] emphasized the need for effective marketing elements, including a strong brand, for firms to be identified and distinguished from others. This underscores the importance of brand image as a key differentiator. A prior study [86] confirmed that consumers predominantly associate with product attributes such as name, packaging, and label. Thus, building a strong brand image is essential to create a positive association with consumers. In support of the hypothesis, [88, 89] provides compelling evidence of the positive impact of brand image on overall firm performance. This establishes a precedent for understanding the broader implications of a strong brand image [89]. Their study revealed that internal branding positively influences start-up brands and performance, emphasizing the need for a cohesive brand identity within the start-up environment.

Lastly, the study also considered establishing the relationship between innovation capabilities and brand image. The findings of the proposed hypothesis were supported and therefore confirm the works of [94]. According to a study by Amoah et al. [128], innovation has a significant impact on brand image hence affirming a positive relationship between innovation capabilities and the perception of brand image. Innovation capabilities empower companies to create distinct processes and brands, ultimately aiming to achieve a competitive advantage. For instance, Halliday and Trott [127] concluded that there is a positive relationship between innovation capabilities and brand image. Thus, companies can enhance their brand image by focusing on innovation mechanisms or managing design innovation effectively. [86] reinforced that firms with innovation capabilities create distinctive processes and products that enhance their brand image. For instance, [86], and Amoah et al., [128] concluded that high levels of innovation are beneficial for brand-focused in building a powerful brand image. Additionally, companies that prioritize technological and design innovation management significantly improve their brand image, further reinforcing a positive link between innovation capabilities and brand image. As indicated by [95] fostering innovation do not only

differentiate a company's products and services but also strengthens its brand image in the competitive market landscape.

## Conclusion

The current study examined the impact of social media usage on the performance of startup in Ghana, mediated by brand image. Drawing upon the theory of resource-based view (RBV), the research formulated hypotheses designed to provide answers to the core objectives. Besides, the study gathered data from young entrepreneurs in Ghana, utilizing Smart PLS software for the analysis. All the five hypotheses developed were supported in the current study. This study establishes that social media marketing and brand image are pivotal marketing tools, that have significant and direct effect on the startup performance. The study also tested the relationship between social media usage and startup performance, mediated by the brand image. The results emerged shows that social media usage allows startups to reach potential customers, enhance brand image, create value and, ultimately, improve business performance. This implies that the positive effect can be attributed to the effective utilization of social media marketing applications, resulting in a larger customer base, enhanced relationships, and better performance. The present study further highlights the importance of innovation capability in improving firm performance, revealing a significant effect on start-up performance.

This research adds to the ongoing conversation on startup utilization of digital technologies, especially social media, to enhance business growth and achieve sustainable performance in the context of developing countries since few empirical studies have been devoted to this context. Moreover, the present study provides a strong understanding of brand image in the relationship between social media usage and start-up performance, witnessing a mediating role of brand image in the relationship between social media usage and start-up performance. Again, the study demonstrates that deploying social media as a resource in an organization tends to impact positively on start-up performance. Finally, this study offers comprehensive insights for young entrepreneurs, small and medium enterprise managers, and policymakers pursuing to improve firm performance through innovation.

### Theoretical implications

The current study offers theoretical contributions. The study first provides a comprehensive insight into understanding social media marketing tools and how start-ups can holistically integrate their marketing strategies. Thus, the present study adds to the existing knowledge regarding the relationship between social media usage and start-up performance in emerging countries. Again, the current study offers a strong insight into the crucial role of the mediating mechanism of brand image and the effect of innovation capability from the start-up's perspective, which is another novelty.

More importantly, the present study broadens the theory applied concerning the effect of innovation capability, offering the newest support by providing empirical support regarding the innovation capability effect on start-ups' performance. Additionally, the study empirically supports the resource-based theory in comprehensively assessing the importance of the innovative strategic tool (i.e., social media marketing) capability on start-up performance. Thus, this study augments explicitly the innovation capability literature by examining social media usage within start-up settings by arguing that in times of brand image influences firm performance. The study further highlighted the effect of brand image in improving start-ups' performance and its mediating effect in reinforcing social media usage and start-up performance relationships in developing countries.

The current study again supplements the existing literature on social media usage and start-ups since the outcome confirms that social media usage directly affects the start-up's

performance. Thus, the study outcomes present a profound understanding of how social media usage enables start-ups to build brand image and attract audience and investors, thereby increasing sales and profitability. Furthermore, this study provided empirical evidence of the importance of social media utilization for start-ups' growth and sustainability. Henceforth, the present study has significant implications for young entrepreneurs and policymakers who intend to achieve sustainable growth.

## Practical/Managerial implications

Practically, this study provides several implications for entrepreneurs and emerging businesses. The current study offers a holistic understanding of social media applications, allowing young entrepreneurs to acknowledge the benefits of social media adoption and utilization. Thus, this study advocates that start-up young entrepreneurs utilize suitable social media applications to enhance business performance by reaching out to the target audience, building relationships among members of the start-up ecosystem, and increasing customer loyalty. For instance, the study outcome demonstrates that social media usage positively impacts start-up performance. It indicates that start-up owners should prioritize and integrate social media technologies into their marketing strategies. Innovation Capability analysis shows that technology as an aspect of innovation capability suggests a high possibility of achieving better financial performance. Therefore, young entrepreneurs and small business owners can focus on different aspects of innovation capability, especially technology (i.e., social media technologies), to increase efficiency and productivity.

Moreover, the present study offers an innovative competitive strategy for start-ups to leverage social media technologies to strengthen managerial decisions to achieve sustenance. Thus, the study provides a framework for supporting sustainable start-ups by giving them broad, innovative strategies to increase brand awareness and sales. Given the need for small business owners to compete and keep up with globalization, the outcome of the study demonstrated that start-ups can quickly respond to a business-changing environment and gain a competitive edge by implementing digital innovative tools, which provide a variety of platforms for enhancing start-ups performance and fostering sustainable growth. Finally, the results of this study may be helpful and serve as a preliminary guide for scholars who wish to investigate social media implementation and start-ups in the context of developing economies.

## Marketing/Logistics implications

Based on the journal's focus, it is essential to acknowledge the influence of social media usage in improving start-up business operations and performance and how it affects marketing and logistics. Social media technologies are an effective marketing tool to foster customer relationships and achieve sustainable growth. Small and medium enterprises, including start-ups, adopting and utilizing social media technologies can strategically help engage customers, provide real-time customer experiences, and offer avenues for engaging stakeholders.

Additionally, it has been shown that social media is a practical, innovative tool that helps start-ups expand their customer base through attraction and creating value, hence driving sustainable business growth. This implies that start-ups can focus on using social media technologies to build a business presence, enhance brand image, and promote strategic positioning. From the organizational perspective, social media marketing should be holistically integrated into their marketing and logistics strategies, emphasizing how they can utilize social media to promote sustainable performance, advance innovative business procedures, and achieve competitive advantage.

### Limitations and future research directions

The limitations of this current study offer opportunities for future research. First, the study gathered data (sample) from a single emerging country, making generalizing the findings difficult since there is a possibility of bias. Therefore, future studies could examine start-ups from different sectors and developed countries using similar constructs to validate the current study outcomes.

Second, our study did not consider any moderating variable relative to social media usage and start-up performance. Hence, researchers could explore other moderating variables in the social media usage and start-up performance relationship in the context of developing countries. Third, the present study investigates the impact of social media usage, considering innovation capability. However, future studies could centre on other innovation capability factors (i.e., organizational intelligence, entrepreneurial orientation, etc.) regarding start-up performance.

Finally, while the study offers valuable insights into the application and benefits of social media in the SME context, the consequences were not highlighted. Researchers could explore social media's negative effect on the business activities and processes of start-ups in developing economies.

### Author contributions

**Conceptualization:** Emmanuel Bruce, Sulemana Bankuoru Egala, Bernard Ekow Baidoo.

**Formal analysis:** Emmanuel Bruce, John Amoah, Philip Adu Sarfo.

**Investigation:** Emmanuel Bruce.

**Methodology:** John Amoah, Sulemana Bankuoru Egala.

**Project administration:** Zhao Shurong.

**Resources:** Dennis Akomanyi Darko.

**Software:** John Amoah.

**Supervision:** Zhao Shurong, Sulemana Bankuoru Egala.

**Validation:** Zhao Shurong, Philip Adu Sarfo.

**Visualization:** Philip Adu Sarfo.

**Writing – original draft:** Emmanuel Bruce, Sulemana Bankuoru Egala, Bernard Ekow Baidoo.

**Writing – review & editing:** Zhao Shurong, Dennis Akomanyi Darko.

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
