## [Decision Letter · Decision Letter 0]

18 Nov 2024

PONE-D-24-29006Examining the Impact of Social Media Usage on Start-ups Performance: Mediating Role of Brand Image.PLOS ONE

Dear Dr. Bruce,

Thank you for submitting your manuscript to PLOS ONE. After careful consideration, we feel that it has merit but does not fully meet PLOS ONE’s publication criteria as it currently stands. Therefore, we invite you to submit a revised version of the manuscript that addresses the points raised during the review process.

We look forward to receiving your revised manuscript.

Kind regards,

Bo Pu, Ph.D.

Academic Editor

PLOS ONE

2. Please ensure that you include a title page within your main document. You should list all authors and all affiliations as per our author instructions and clearly indicate the corresponding author.

Additional Editor Comments:

this manuscript should be improved.

Reviewers' comments:

Reviewer's Responses to Questions

**Comments to the Author**

1. Is the manuscript technically sound, and do the data support the conclusions?

Reviewer #1: Yes

Reviewer #2: Yes

2. Has the statistical analysis been performed appropriately and rigorously? 

Reviewer #1: Yes

Reviewer #2: Yes

3. Have the authors made all data underlying the findings in their manuscript fully available?

Reviewer #1: Yes

Reviewer #2: Yes

4. Is the manuscript presented in an intelligible fashion and written in standard English?

Reviewer #1: Yes

Reviewer #2: Yes

5. Review Comments to the Author

Reviewer #1: The manuscript showed a technically sound paper of scientific research with data that supports the conclusions. The paper aimed to explore the impact of social media usage to startup performance with a mediating role of brand image. Data collected by sending the questionnaires to startup firms in Ghana and have 450 respondents in valid. The paper show the respondents profile, conducted the factor loading, Cronbach's Alpha, CR, VIF, AVE and Fornell-Larcker for measurement variables; hypothesis testing by PLS-SEM. The discussions, conclusions and implications were drawn appropriately based on the data collected.

The statistical analysis was performed approciately and rigorously.

Data collected are permitted from National Board for Small-Scale Industries (NBSSI) and the respondents agreed to participating. However, the data was not divided into the direct responses and google form responses. The paper has not showed how to choose the start-ups to sending the questionnaire. So that, the authors should be show the data more clearly in this points.

The manuscript was showed in intelligible fashion and written in standard English. However, the paper should considerably the form of citation. The citation based on the number of references made the target readers to have to check "what is it", eg. line 35 "From a start-up perspective, [15] noted...." the readers don't know what is [15]; or line 40 "According to [17],..." what is [17],... there have a lot of above-mentioned citations.

The paper is too long, the authors should be rewrite for shorter.

Reviewer #2: I consider the article very interesting and solid . A few adjustments to the language and the text should be made. For example:

59-60 It is not clear what you wanted to say. Seems distinct from the aim stated in the title and the rest of the paper. And probably would sound better with ”the relationship between ...”

382 - Indicator 1 of construct IC is not very clear.

382 - Construct SP indicators should start from 1.

6. PLOS authors have the option to publish the peer review history of their article (what does this mean? ). If published, this will include your full peer review and any attached files.

**Do you want your identity to be public for this peer review?** For information about this choice, including consent withdrawal, please see our Privacy Policy .

Reviewer #1: **Yes: ** Trung Bao

Reviewer #2: No

---

## [Author Response · Author response to Decision Letter 0]

22 Jan 2025

Comments from Reviewer 1

Comments and Suggestions for Authors

Thank you for these observations. We have rewritten the manuscript to meet the journal style requirement.

2. Please ensure that you include a title page within your main document. You should list all authors and all affiliations as per our author instructions and clearly indicate the corresponding author.

Thanks for the comment. We agree with the reviewer and have added the title page in the manuscript, listing all the authors and their affiliations.

Emmanuel Bruce1*, Zhao Shurong 2, John Amoah3, Sulemana Bankuoru Egala 4, Philip Adu Sarfo5 Dennis Akomanyi Darko 1

1 School of Management and Economics, University of Electronic Science and Technology of China, Chengdu 611731, China; Center for West Africa Studies, UESTC, Chengdu, China, 2 School of Public Affairs and Administration, University of Electronic Science and Technology of China, Chengdu 611731, China; Center for West Africa Studies, UESTC, Chengdu, China, 3 John Amoah, Department of Marketing and Strategy, Takoradi Technical University, Takoradi, Ghana, 4 Department of Informatics, Faculty of ICT, SD Dombo University of Business and Integrated Development Studies, Wa, UW/R, Bamahu, Ghana, 5 School of Management Engineering, Zhengzhou University, 45001, Zhengzhou, China

kinbuki100@outlook.com*

We thank you for the comments ones again. Respectively, we have provided additional references at the references at the last page of the manuscript, reference number 127, 128, as seen as follows;

127. Halliday, S.V. and Trott, P., Relational, interactive service innovation: building branding competence. Marketing Theory, 2010, 10(2), pp. 144-160.

128. Amoah, J., Jibril A. B., Bankuoru Egala, S., and Keelson, S. A. (2022). Online brand community and consumer brand trust: Analysis from Czech millennials. Cogent Business & Management, 2022, 9(1), 2149152.

1. Is the manuscript technically sound, and do the data support the conclusions?

We thank you for the comments. Respectively, we are unable to conduct emperiment as suggested. However, we have rewritten the conclusion based on the data analysis, presented in the manuscript at the conclusion part, as follows:

The current study examined the impact of social media usage on the performance of startup in Ghana, mediated by brand image. Drawing upon the theory of resource-based view (RBV), the research formulated hypotheses designed to provide answers to the core objectives. Besides, the study gathered data from young entrepreneurs in Ghana, utilizing Smart PLS software for the analysis. All the five hypotheses developed were supported in the current study. This study establishes that social media marketing and brand image are pivotal marketing tools, that have significant and direct effect on the startup performance. The study also tested the relationship between social media usage and startup performance, mediated by the brand image. The results emerged shows that social media usage allows startups to reach potential customers, enhance brand image, create value and, ultimately, improve business performance. This implies that the positive effect can be attributed to the effective utilization of social media marketing applications, resulting in a larger customer base, enhanced relationships, and better performanc

e.

The present study further highlights the importance of innovation capability in improving firm performance, revealing a significant effect on start-up performance. This research adds to the ongoing conversation on startup utilization of digital technologies, especially social media, to enhance business growth and achieve sustainable performance in the context of developing countries since few empirical studies have been devoted to this context.

Reviewer #1: The manuscript showed a technically sound paper of scientific research with data that supports the conclusions. The paper aimed to explore the impact of social media usage to startup performance with a mediating role of brand image. Data collected by sending the questionnaires to startup firms in Ghana and have 450 respondents in valid. The paper show the respondents profile, conducted the factor loading, Cronbach's Alpha, CR, VIF, AVE and Fornell-Larcker for measurement variables; hypothesis testing by PLS-SEM. The discussions, conclusions and implications were drawn appropriately based on the data collected.

The statistical analysis was performed approciately and rigorously.

Data collected are permitted from National Board for Small-Scale Industries (NBSSI) and the respondents agreed to participating. However, the data was not divided into the direct responses and google form responses. The paper has not showed how to choose the start-ups to sending the questionnaire. So that, the authors should be show the data more clearly in this points.

The manuscript was showed in intelligible fashion and written in standard English. However, the paper should considerably the form of citation. The citation based on the number of references made the target readers to have to check "what is it", eg. line 35 "From a start-up perspective, [15] noted...." the readers don't know what is [15]; or line 40 "According to [17],..." what is [17],... there have a lot of above-mentioned citations.

The paper is too long, the authors should be rewrite for shorter.

Reviewer #2: I consider the article very interesting and solid. A few adjustments to the language and the text should be made. For example:

59-60 It is not clear what you wanted to say. Seems distinct from the aim stated in the title and the rest of the paper. And probably would sound better with ”the relationship between ...”

Thanks for the comment. We agree with the reviewer and have explained how direct responses and google form were filled by the responses, as shown in the manuscript (methodology section)

“Out of the 520 questionnaires administered, 450 of the questionnaires received were valid for data processing and analysis. The questionnaires were administered via Goggle Forms (i.e. through emails and social media platforms such as Facebook Messenger and WhatsApp) and offline methods after formal approval from the selected young entrepreneurs. Additionally, young entrepreneurs were chosen based on those who utilizes social media networks for their business activities and as a marketing communication tool”.

Again, issues related to citations in the manuscript has been resolved and rewritten to sound well.

For citation number 15 for instance “As summarized by [15-17] for instance, young entrepreneurs have started utilizing social media to facilitate effective communication with consumers and business visibility. As a result, social media adoption can lead to double profits, which boosts marketing effectiveness [4].

For citation number 35 “In investigating social media usage and young entrepreneur investment, Chen and Liu [35] revealed how social media has become an important marketing tool for entrepreneurship activities.

For citation number 40 “Moreover, other researchers [31, 35, 40] have employed the research-based view theory to investigate the impact of social media on start-up performance and sustainability.

For citation number 59 and 60 have been rewritten “Okrah et al. [57] focus on factors of startup success and growth by arguing that social media technologies are taking over the Ghanaian business landscape to drive sustenance, especially among small and medium enterprises. Several works have established that social media enables startups in Ghana to connect the entire ecosystem [59, 60], sharing knowledge [61] and networking [62].

Besides, the overall manuscript has been rewritten to make it shorter as requested.

Furthermore, issues related with 382 - Indicator 1 of construct IC and Construct SP indicators, the author have tackled this issue, dwelling on innovation capabilities and brand image, presented in the manuscript (literature review), developing hypothesis for it and removed all the notes on mediating role of brand image. Besides, the authors have provided discussion for the relationship between innovation capabilities and brand image in the manuscripts “Lastly, the study also considered establishing the relationship between innovation capabilities and brand image. The findings of the proposed hypothesis were supported and therefore confirm the works of [94]. According to a study by Amoah et al. [128], innovation has a significant impact on brand image hence affirming a positive relationship between innovation capabilities and the perception of brand image. Innovation capabilities empower companies to create distinct processes and brands, ultimately aiming to achieve a competitive advantage. For instance, Halliday and Trott [127] concluded that there is a positive relationship between innovation capabilities and brand image. Thus, companies can enhance their brand image by focusing on innovation mechanisms or managing design innovation effectively. [86] reinforced that firms with innovation capabilities create distinctive processes and products that enhance their brand image. For instance, [86], and Amoah et al., [128] concluded that high levels of innovation are beneficial for brand-focused in building a powerful brand image. Additionally, companies that prioritize technological and design innovation management significantly improve their brand image, further reinforcing a positive link between innovation capabilities and brand image. As indicated by [95] fostering innovation do not only differentiate a company’s products and services but also strengthens its brand image in the competitive market landscape”.

---

## [Decision Letter · Decision Letter 1]

14 Feb 2025

Examining the Impact of Social Media Usage on Start-ups Performance: Mediating Role of Brand Image.

PONE-D-24-29006R1

Dear Dr. Bruce,

We’re pleased to inform you that your manuscript has been judged scientifically suitable for publication and will be formally accepted for publication once it meets all outstanding technical requirements.

Kind regards,

Bo Pu, Ph.D.

Academic Editor

PLOS ONE

Additional Editor Comments (optional):

Thanks for your improvement towards this manuscript.

Reviewers' comments:

Reviewer's Responses to Questions

**Comments to the Author**

1. If the authors have adequately addressed your comments raised in a previous round of review and you feel that this manuscript is now acceptable for publication, you may indicate that here to bypass the “Comments to the Author” section, enter your conflict of interest statement in the “Confidential to Editor” section, and submit your "Accept" recommendation.

Reviewer #1: All comments have been addressed

2. Is the manuscript technically sound, and do the data support the conclusions?

Reviewer #1: Yes

3. Has the statistical analysis been performed appropriately and rigorously? 

Reviewer #1: Yes

4. Have the authors made all data underlying the findings in their manuscript fully available?

Reviewer #1: Yes

5. Is the manuscript presented in an intelligible fashion and written in standard English?

Reviewer #1: Yes

6. Review Comments to the Author

Reviewer #1: 1. Expand the Literature Review: Offer a more comprehensive discussion of the theoretical framework, delving deeper into its relevance and application to the study. Critically evaluate existing studies to identify gaps in the literature and clearly articulate how this research addresses those gaps.

2. Improve Methodological Transparency: Provide detailed information about the survey instrument, including sample items or an appendix with the full questionnaire. Justify the choice of measurement scales, explaining why they are appropriate for the constructs being measured and how they align with the study's objectives.

3. Examine Demographic Data: Analyze the demographic characteristics of respondents in greater depth. Discuss how factors such as gender, education, or industry representation might influence the results and their broader implications for start-ups operating in diverse sectors.

4. Enhance the Discussion Section: Expand on the theoretical contributions of the study, explaining how the findings advance existing frameworks or theories. Provide a more critical analysis of the results, comparing them with prior research and highlighting any unexpected outcomes or novel insights.

5. Overall Assessment: The paper addresses an important and timely topic with a robust methodology and provides valuable insights for both academics and practitioners. However, there are areas for improvement, particularly in the literature review, discussion of limitations, and theoretical contributions. With revisions, the paper has the potential to make a significant contribution to the field of entrepreneurship and social media research.

7. PLOS authors have the option to publish the peer review history of their article (what does this mean? ). If published, this will include your full peer review and any attached files.

**Do you want your identity to be public for this peer review?** For information about this choice, including consent withdrawal, please see our Privacy Policy .

Reviewer #1: No

---

## [Editor Report · Acceptance letter]

PONE-D-24-29006R1

PLOS ONE

Dear Dr. Bruce,

I'm pleased to inform you that your manuscript has been deemed suitable for publication in PLOS ONE. Congratulations! Your manuscript is now being handed over to our production team.

Kind regards,

on behalf of

Dr. Bo Pu

Academic Editor

PLOS ONE